# Masked Autoencoding for Scalable and Generalizable Decision Making

**Fangchen Liu**[1] *   **Hao Liu**[1] *   **Aditya Grover**[2]   **Pieter Abbeel**[1]

[1] Berkeley AI Research, UC Berkeley     [2] UCLA

* Equal contribution

{fangchen_liu, hao.liu}@berkeley.edu

## Abstract

We are interested in learning scalable agents for reinforcement learning that can learn from large-scale, diverse sequential data similar to current large vision and language models. To this end, this paper presents masked decision prediction (MaskDP), a simple and scalable self-supervised pretraining method for reinforcement learning (RL) and behavioral cloning (BC). In our MaskDP approach, we employ a masked autoencoder (MAE) to state-action trajectories, wherein we randomly mask state and action tokens and reconstruct the missing data. By doing so, the model is required to infer masked-out states and actions and extract information about dynamics. We find that masking different proportions of the input sequence significantly helps with learning a better model that generalizes well to multiple downstream tasks. In our empirical study, we find that a MaskDP model gains the capability of zero-shot transfer to new BC tasks, such as single and multiple goal reaching, and it can zero-shot infer skills from a few example transitions. In addition, MaskDP transfers well to offline RL and shows promising scaling behavior w.r.t. to model size. It is amenable to data-efficient finetuning, achieving competitive results with prior methods based on autoregressive pretraining[1].

## 1 Introduction

Self-supervised pretraining has made tremendous successes for unsupervised representation learning in natural language processing (NLP) and vision [12, 9, 3, 4]. These methods work by predicting a removed portion of the data, which is often referred to as masked token prediction. By varying the masking patterns and architectures, different methods have been developed for NLP and vision, $e.g.$, Transformer [33], GPT [4], BERT [9], and MAE [12]. These methods are simple to implement and scalable to large Internet-scale datasets and deep neural networks, leading to excellent flexibility and generalization for downstream tasks [9, 12, 1].

In this work, we explore the generality of masked token prediction for generalizable and flexible reinforcement learning (RL). Prior work has explored sequence modeling for sequential decision making in the context of offline RL $e.g.$ decision transformer (DT) [6] and trajectory transformer (TT) [13], and black-box optimization $e.g.$ transformer neural processes (TNP) [19]. These methods are based on autoregressive next token prediction, similar to GPT [4]. While promising, these works do not leverage diverse unlabeled data for generalization across various downstream tasks. In addition, DT [6] needs reward-labeled high quality datasets, while TT [13] requires discretizing states and actions, further limiting its applicability. The flexibility of applying arbitrary masks for executing various task specifications in RL significantly lags behind NLP and vision.

---

[1]The implementation of MaskDP is available at `https://github.com/FangchenLiu/MaskDP_public`

36th Conference on Neural Information Processing Systems (NeurIPS 2022).

**Figure 1:** Illustration of MaskDP. During pretraining stage, we perform the masked token prediction task. And after pretraining, the model can be deployed to various downstream tasks using different mask patterns.

We propose Masked Decision Prediction (MaskDP), a pretraining method to learn generalizable models that achieve data-efficient adaptation to various downstream tasks. MaskDP is a self-supervised pretraining method that can leverage unlabeled diverse data. With MaskDP pretraining, the model can generalize well to both goal reaching and offline RL, two distinctive and popular RL paradigms.

Our first key observation is that masked token prediction with random masking similar to BERT [9] and MAE [12] provides a general and flexible way for learning from unsupervised data. Unlike autoregressive action prediction used in prior works, random masking is strictly more general and requires the model to infer masked out states and actions, and thus leads to a single model that can reason about both the forward and inverse dynamics from each sample.

Our second key observation is that since states and actions are highly correlated temporally, trajectories have significantly lower information density, $i.e$, it is easier to predict action or state based on nearby states and actions. Consequently, a high mask ratio ($95\%$) is necessary to make reconstruction task meaningful. Unlike in MAE [12] and BERT [9] where the goal is learning representations, we want to directly apply MaskDP to various downstream tasks, and different mask ratios induce different pre-train and downstream gaps. For example, consider the goal-reaching task within certain time limit. Given current state, future goal and mask tokens between them, the model should be able to inpaint intermediate actions as the goal-reaching plan. The mask ratio varies from short-term plans to long-term plans. Therefore, we combine multiple different mask ratios ($e.g.$ $15\%$, $35\%$, $75\%$, and $95\%$), and mask a portion of data using a randomly sampled mask ratio. Our experiments show that doing so is crucial to achieving high performance. We show that self-supervised pretrained MaskDP achieves high performance in challenging multiple goals reaching setting, outperforming strong baselines in a zero-shot manner.

We highlight our key results here:

- **Single goal reaching**: MaskDP achieves performance that exceeds or matches both training from scratch task-specific methods and other pretraining based methods.
- **Sequential multiple goal reaching**: MaskDP can reach a sequence of goals effectively, even without closed-loop execution, while outperforming iterative baselines significantly.
- **Offline RL**: MaskDP achieves competitive results as specialized approaches. Notably, we demonstrate that non-autoregressive architecture works well for offline RL tasks.

## 2   Related work

**Masked modeling in language and vision.**   Large-scale language models are highly successful [9, 4]–after pretraining on a large amount of data, these pretrained representations generalize well to various downstream tasks. Taking inspiration from the success in NLP, Transformer [33] based

methods have been proposed to model images [7, 10, 3, 12]. iGPT [7] operates on sequences of pixels and predicts unknown pixels. BEiT [3] proposes to predict discrete tokens [32, 22]. MAE [12] proposes to randomly mask patches of the input image and reconstruct the missing pixels. Since we apply random mask across states and actions, our work is also related to prior work on masked prediction across multiple input modalities [see e.g. 34].

**Masked modeling in RL**    Masking trajectories for pretraining has been studied in RL [21, 23, 6, 13, 39]. MVP [35] studies transferring pretrained visual representations to RL tasks. MWM [25] studies masked prediction over convolutional features, and learn a latent dynamics model. Modelling inverse dynamics has also been studied for robot learning from demonstrations and sim-to-real transfer [8, 30]. TT [13] studies autoregressive next token prediction for model-based RL applications. DT [6, 39] study masking autoregressive next token prediction conditioned on return. TNP [19] and BONET [14] study autoregressive masking for sequential decision making for black-box optimization. ICM [21] and SPR [23] study predicting masked state and action in a transition tuple for exploration. Different from these works, MaskDP randomly masks a portion of trajectories and generalizes prior masking strategies such as inverse dynamics. In addition, MaskDP generalizes well to downstream tasks while prior work is task-specific. Concurrent to our work, Uni[MASK] [5] proposes using the bidirectional transformer to predict masked states and actions and demonstrates that the resulting model performs well on a large variety of tasks. The main difference between our works is that Uni[MASK] is more interested in comparing the performance between training regimes with different masking schemes, while our work focuses on solving a range of downstream tasks with minimal task-specific designs during pretraining.

**Unsupervised pretraining in RL**    Our work falls under the category of self-supervised pretraining in RL. Self-supervised discovery of a set of task-agnostic behaviors by means of seeking to maximize an intrinsic reward has been explored as intrinsic motivation [2], often with the goal of encouraging exploration [27, 20]. APT [18] studies nonparametric entropy maximization for pretraining and is extended to learning skills [17]. Proto-RL [36] further improves pretraining by representation learning. CIC [16] combines contrastive learning with skill discover and improves results on URL benchmarks [15]. APV [26] shows successful transfer of pretrained representation across domains. Many of these methods are used to pretrain agents that are later adapted to specific reinforcement learning tasks. Using offline data for pretraining agents has also been explored in prior work [24, 28, 38]. SGI [24] proposes combining self-predictive representation [23] and inverse dynamics prediction. ATC [28] studies contrastive pretraining on trajectories and shows transferring the representations to downstream tasks.

## 3   Method

The key idea in MaskDP is to mask and reconstruct state-action sequences during the pretraining stage. Post pretraining, MaskDP can be zero-shot deployed or finetuned for various downstream tasks. The paradigm of the model for pretraining and finetuning is summarized in Figure 1.

### 3.1   MaskDP Pretraining

**Random masking.**    For sequences with low information density, a *high* masking ratio is typically applied to eliminate information redundancy and make the task sufficiently difficult to avoid trivial interpolation from visible neighbor tokens. However, unlike vision and language, where the goal is to learn good representations; we also consider directly deploying this model by leveraging its inpainting capability for various downstream tasks. For example, we can give the model a goal at timestep $T$ and mask all the future inputs, the model can generate intermediate actions by inpainting the mask tokens. The mask ratio varies from goal to goal, depending on the time budget. To reduce the gap between training and deployment, we keep a set of mask ratios (*i.e.* $15\%$, $35\%$, $50\%$, $75\%$, and $95\%$), and the data is randomly masked with a ratio sampled from this set. We find that masking multiple proportions of the input yields a meaningful self-supervisory task.

We apply random masking on state tokens and action tokens independently. By doing so, the model is implicitly learning both forward and inverse dynamics. This also provides more flexibility as we can provide state or action-level inputs but not transition-level.

**Architecture**    Our encoder is a Transformer [33] but applied only on visible, unmasked states and actions, similar to MAE [12]. The states and actions are first embedded by separated linear layers, positional embeddings are then added, and lastly, the embeddings are processed by a series of self-attentional blocks. The decoder operates on the full set of encoded visible state and action tokens and mask tokens. Each mask token is a shared, learned vector that indicates the presence of a missing token to be predicted. Similar to the encoder, the masked whole sequence will pass through separated linear projections added with positional embedding prior to being passed to the decoder. Both the encoder and decoder are bidirectional.

**Prediction target**    Our MaskDP reconstructs the input by predicting the whole action and state sequences. The last layer of the decoder consists of two MLPs to decode states and actions separately. The loss function computes the mean squared error (MSE) between the reconstructed whole sequence and original inputs. Different from other masked prediction variants [12, 9], we found mask loss is not useful in our setting, as our goal is to obtain an scalable decision making model but not only for representation learning.

### 3.2  MaskDP Downstream Tasks

**MaskDP for goal reaching**    We consider the problem of reaching one goal or multiple goals from a given state. The model has to generate a sequence of actions to reach goals within a certain amount of steps. MaskDP denoising pretraining objective fits the goal reaching scenario well as the model must learn to inpaint masked actions based on remaining states. In this task, the MaskDP encoder input is a concatenation of initial state and goals, and the decoder input is a concatenation of initial state embedding, masked token sequence, and goal embeddings. Note that the number of masked tokens determines the number of timesteps the model is expected to reach the given goals. The model then generates a state-action sequence, where we can directly execute the whole action sequence (namely "open-loop"), or only execute the first action and forward the model again with the obtained new observation (namely "closed-loop").

**MaskDP for skill prompting**    Skill prompting requires the model to generate a trajectory conditioned a given context. For example, consider a walker agent: if we prompt it with a few state-action pairs of walking/running/standing, it should continue to generate a trajectory in the same skill pattern. Accordingly, we append the observed initial state-action sequences with masked tokens for the future. The model can be rolled out once to generate the whole future sequence, or queried repeatedly to refill the masked tokens at each time step. Similar to goal-reaching task, we refer to these strategies as "open-loop" and "closed-loop" respectively.

**MaskDP for offline RL**    In offline RL, the objective is to learn one model for maximizing the return for a task specified by a reward function. This is different from our self-supervised pretraining target, so extra finetuning is needed. We adopt a standard actor-critic framework similar to TD3 [11] by adding a critic head and actor head, where the actor takes a state sequence as input, and the critic takes the state-action sequence as input. Both are mask-free. To match the setting in RL, we change the bidirectional attention mask in the transformer to a causal attention mask. More details about RL finetuning can be found in section 4.2.3.

## 4  Experiments

In our experiments, we evaluate transfer learning in downstream tasks using MaskDP. Section 4.1 introduces the environments, pretraining, and the baselines compared in experiments. Section 4.2 summarizes the results of MaskDP on goal reaching, skill prompting, and offline RL. Through further analysis in Section 4.3, we present an ablation study on various design choices of our model.

### 4.1  Experiments Setup

**Environments: domains vs. tasks**    We adopt the environment setup used in EXoRL [37], based on DeepMind control suite [29], where a domain describes the type of agent (e.g. Walker) but tasks are specified by rewards (e.g., Walker walk, Walker run). We use 3 domains (Walker, Cheetah and Quadruped) with 7 tasks in total. More details about the environments can be found in the Appendix.

**Pretraining datasets**   Real-world pretraining data generally varies greatly in quality. To mimic this, we construct two different pretraining datasets to approximate different data quality scenarios.

- **Near-expert**: For every task, we train a TD3 agent [11] for 1M steps and freeze its parameters. We rollout the policy with Gaussian random noise and collect 4M experience on each task.
- **Mixed**: This dataset consists of diverse data collected from various agents, including 2K near-expert trajectories for each task. Similar to ExoRL [37], we collect 10M exploratory trajectories using intrinsic reward from Proto-RL [36] for each domain. We also to use a TD3 [11] agent to maximize the sum of extrinsic reward and the Proto-RL intrinsic reward, and store its 2M experience on each task.

For more details about the above datasets and more ablations on the dataset quality, please refer to Section A and Section B.1 respectively. We perform both single-task and multi-task pretraining using the above datasets. The former leverages task-specific data while the latter utilizes data from all tasks within the same domain. We pretrain agents for 400K gradient steps. Specifically, for the model pretrained on the near-expert dataset, we perform zero-shot[2] evaluation of goal reaching and skill prompting, and finetuning for offline RL; for model trained on the mixed dataset, we provide the finetuning results in Section 4.3 and Section A.

**Baselines**

- GPT. We train an autoregressive model similar to GPT [4] which takes the past states and actions as input to predict the next state or action.
- Goal-GPT. We specifically modify GPT to Goal-GPT to evaluate its performance on goal reaching tasks. The model takes current goal and observations as input, and predicts the action to reach this goal. The model is trained using a behaviour cloning loss as [6].
- Goal-MLP. Standard behavior cloning method that conditions on the goal. The major difference between this and Goal-GPT is here we do not use the causal Transformer architecture to make the history visible.

By default, MaskDP uses a 3-layer encoder and 2-layer decoder, and the baselines based on GPT use 5 attention layers. MaskDP and all the above models are comparable with similar architecture design and size, and share the same training hyper-parameters. Details about the architecture and training of MaskDP and the above baselines can be found in Section A.

### 4.2   Main Results

#### 4.2.1   Goal Reaching

We consider both single and multiple goal-reaching settings. The agent is required to reach one or multiple goals from a given state, which are all sampled from the same trajectory to guarantee reachabilty within a reasonable time budget. During evaluation, the agent rolls out to reach the given goal(s) within a time budget. The evaluation dataset is also collected by the same RL agent in 3 environments with different seeds, which is unseen during pretraining. The detailed settings are:

- **single-goal reaching**: For every trajectory in the validation set, we randomly sample a start state and a future state in $T \in [15, 20)$ steps as the goal. All the methods are evaluated on the same set of 300 state-goal pairs with a given budget of $T + 3$. We set the agent to the start state and report the L2 distance between the goal and the closest rollout state within this budget.
- **multi-goal reaching**: For every trajectory in the validation set, we randomly sample a start state and 5 goal states at random future timesteps from $[12, 60)$. We evaluate the same set of 100 state-goal sequences and add additional 5 timestep budgets for all the goals. Similar to single-goal reaching, We report the L2 distance between every goal and the closest rollout state before running out of its corresponding budget.

We show the zero-shot performance of MaskDP and baselines pretrained with the near-expert data (both in single-task and multi-task settings). We report L2 distance averaged over the states and goals

---

[2]We directly evaluate the model on some unseen state-goal pairs in the validation dataset

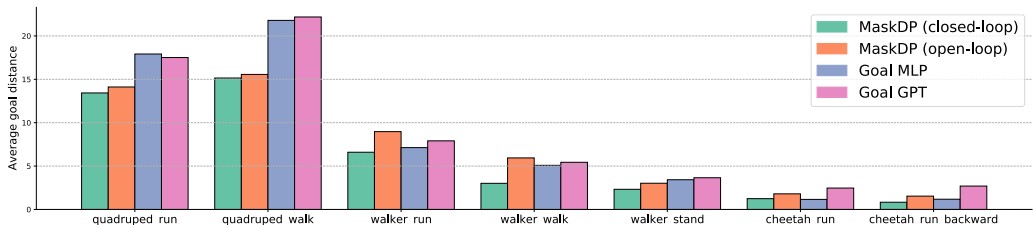

**Figure 2:** Single task pretraining followed by single goal reaching downstream task. MaskDP with closed-loop execution achieves the best performance on all the tasks, and get the most significant improvements in the Quadruped domain, which is higher dimensional.

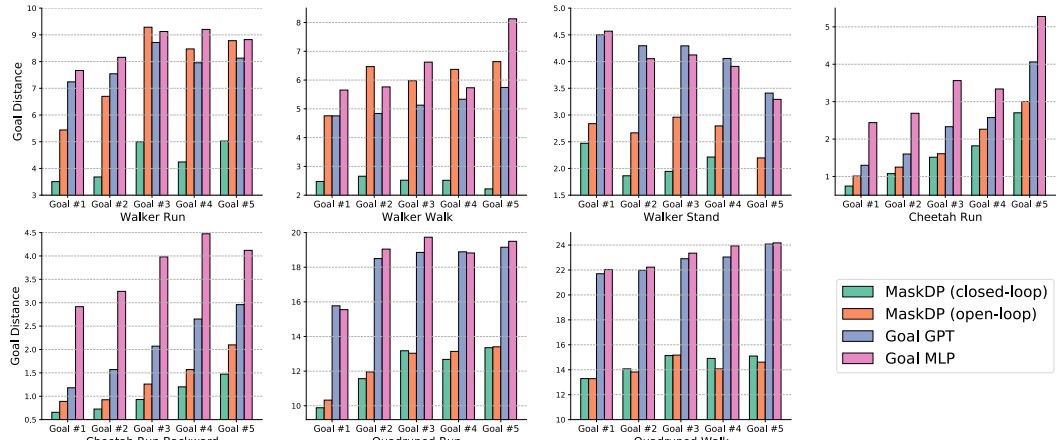

**Figure 3:** Single task pretraining followed by multiple goals reaching downstream task. MaskDP achieves significant improvement on all the tasks with better flexibility in sequential goal reaching.

sampled based on the above rules. Tables of run numbers and standard derivations can be found in Section C.

Figure 2 show the results of reaching single goal. The y-axis is the L2 distance (the lower the better). We observe that both MaskDP (open-loop) and MaskDP (closed-loop) outperform Goal-GPT and Goal-MLP. Despite Goal-GPT being a natural formulation for goal reaching, MaskDP reaches a lower distance to the goal. We attribute the effectiveness to learning a better understanding of the forward and inverse dynamics implicitly. We also observe that the advantages of MaskDP are even more significant in higher dimensional environments, such as Quadruped.

For the more challenging multi-goal reaching task, MaskDP has a significant advantage in flexibility: we can just provide the goals at specific time budgets with interleaved masks and get an executable plan; however, for Goal-MLP and Goal-GPT, we have to change goals at certain timesteps to fulfill future multiple goals. As shown in Figure 3, MaskDP outperforms both goal-GPT and BC by a large margin. In Figure 14, we showed that having "foresight" about future goals can help the agent to generate a better plan.

We can get similar conclusions from the multi-task pretrained models in Figure 4 and Figure 5, where our method consistently works well on all domains, with the most visible advantage in multi-goal reaching setup.

### 4.2.2 Skill Prompting

We are interested in the learned behavior of pretrained models. We use prompting, which has become popular in analyzing models ever since GPT [4]. To do so, we give the agent a short state-action segment randomly cropped from an expert trajectory, set the agent to the last state of the segment, and let the model continue to generate consecutive behaviors. We evaluate the quality of the generated sequence by comparing its obtained rewards with the rollout of a skilled expert.

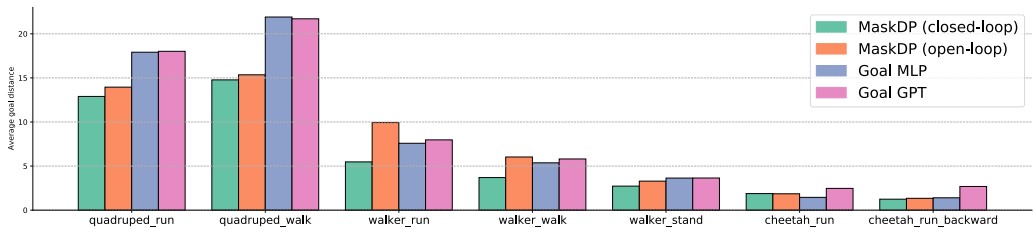

**Figure 4:** Multiple tasks pretraining followed by single goal reaching downstream task, where MaskDP with closed-loop execution works the best, especially in the Quadruped domain.

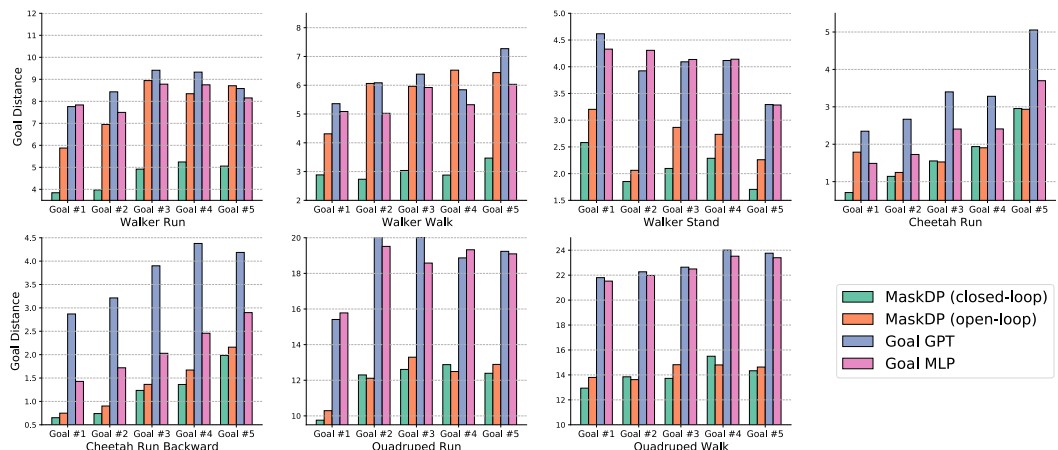

**Figure 5:** Multiple task pretraining followed by multiple goals reaching downstream task.

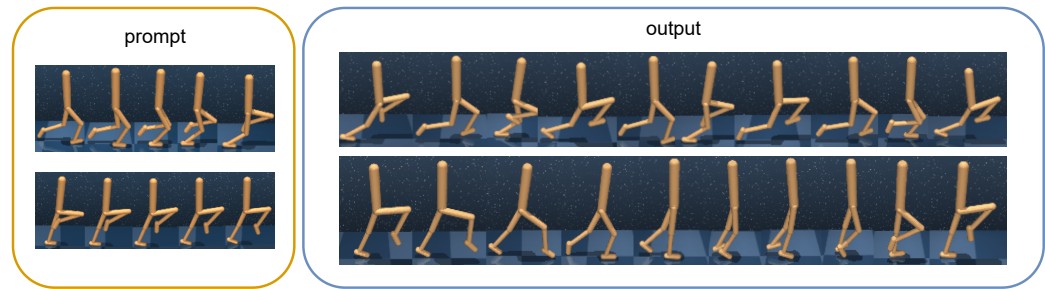

**Figure 6:** Qualitative results on for skill prompting in the Walker domain. Given 5 initial states, the model learns to forecast future trajectories as in the expert-level behaviour.

To be specific, we prompt the multi-task model trained with expert data and sample a 5-timestep state-action segment from $T \in [100, 900]$, where the agent can be walking/running at low or high speeds. We prompt the model with this short segment and let the model generate rollouts for 20, 40, 60 timesteps. We provide both qualitative results and quantitative results in Figure 7 and Figure 6 respectively. We can see in Figure 7, both our method and GPT can match the expert return. It shows that our method can perform as well as autoregressive model in generation task.

### 4.2.3 Offline Reinforcement Learning

**Evaluation** We provide a 2M buffer of the data collected by Proto-RL [36] as in ExoRL [37] does, where the overall return of the data is quite low and thus the BC-based method cannot work well. ExoRL [37] simply shows that an offline TD3 agent works the best on diverse low-return offline data.

We can modify MaskDP to this setting by adding additional actor and critic heads on top of the pretrained encoder, and performing RL training. We evaluate the efficiency of the pretrained model by its return after certain TD gradient steps. The results are shown in Figure 8 averaged over 3

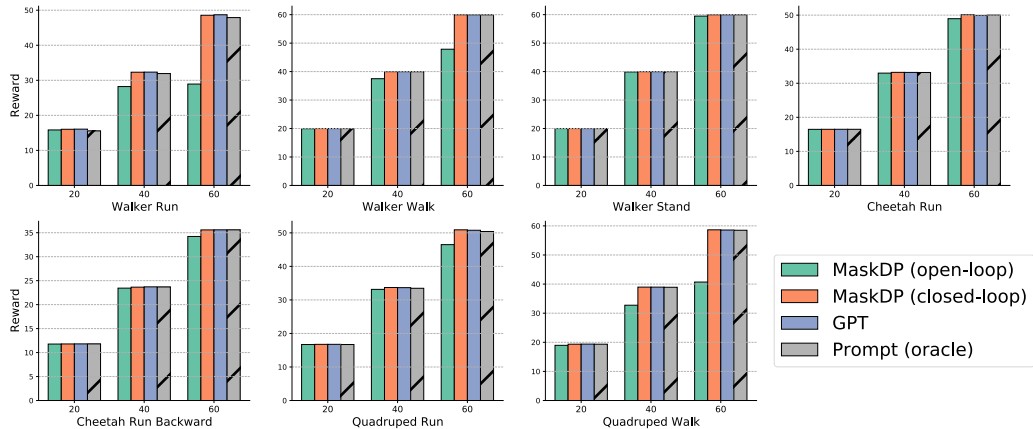

**Figure 7:** Quantitative results on learned behaviors using prompt. Both MaskDP and GPT can match or even slightly surpass the expert-level performance (right grey bar) in trajectory forecasting.

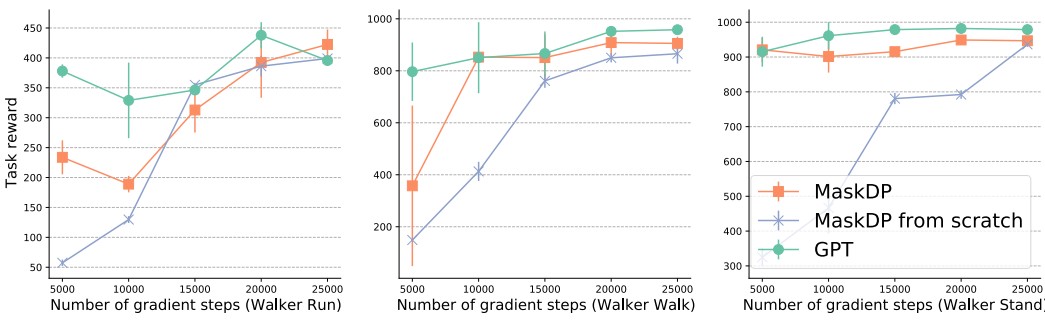

**Figure 8:** Offline RL results on Walker domain. The result of MaskDP matches the GPT-style pretraining performance, and are all comparable to the SoTA in ExoRL [37].

seeds. We observe MaskDP is capable of adapting to downstream tasks quickly, outperforms training from scratch, and achieves similar results as the GPT baseline. Note that in this setting, we need to replace the bidirectional attention mask with the causal attention mask, so there is a larger gap between pretraining and downstream tasks finetuning compared with GPT, which is trained with causal masking. Note that MaskDP from scratch is almost the same as GPT from scratch (both with causal masking). From Figure 8, both MaskDP and GPT can match the best result in ExoRL [37] from their offline TD3 agent, where BC-based method cannot successfully solve this task.

### 4.3 Analysis

**Model scalability.** We also pretrain our agent and baselines on the diverse "mixed" dataset. We compare the smaller version of MaskDP and Goal-GPT[3] on the Quadruped goal reaching problem as shown in Figure 9. The x-axis is the finetuning gradient steps on the expert dataset, and y-axis is the L2 distance to the goal (the lower

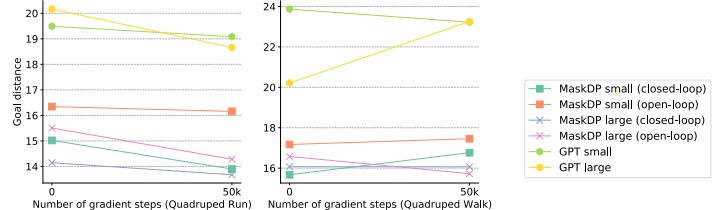

**Figure 9:** Model scalability on two tasks. X-axis represents number of gradient steps. With MaskDP pretraining, larger models outperform smaller models.

---

[3]Both use 3 attention layers.

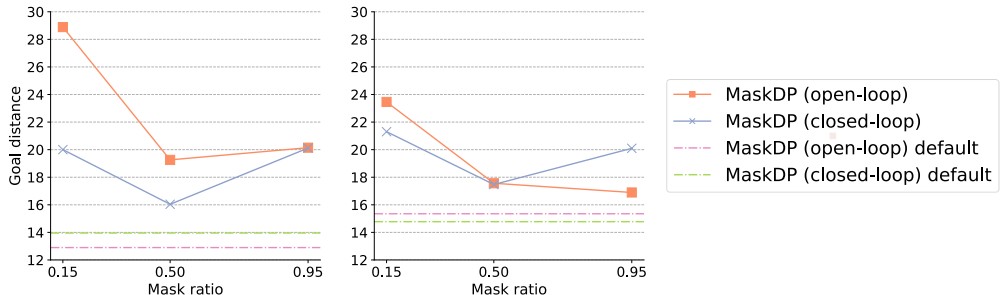

**Figure 10:** Mask ratio ablation. We compare our multiple ratio pretrained model with models trained with fixed ratios, where our masking strategy can achieve much better performance.

the better). We found for both zero-shot evaluation and finetuning, our model's performance improves when model size is enlarged, whereas for Goal-GPT the performance gain is not obvious.

**Mask ratio ablation.** Figure 10 shows the influence of the masking ratio. With a fixed mask ratio, we observe that an extremely high mask ratio (95%) generally does not work well and the typical mask ratio (15%) used in BERT seems to perform much worse than others. A middle mask ratio 50% performs reasonably well, despite still being surprisingly high, similiar to the observations in MAE. However, our mixed mask ratio strategy strictly outperforms all the above options.

**Predicting unmasked tokens ablation.** We also compare the model trained with mask loss vs. total loss. As shown in Figure 11, empirically we do not find mask loss has more advantages than total loss, even on the relatively clean expert dataset, it converges slower than using total loss. For the results on diverse data, please refer to Section C.

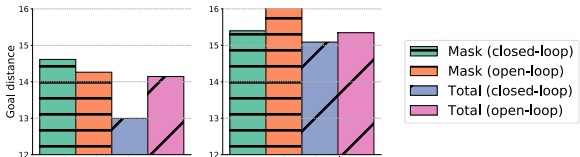

**Figure 11:** Masked loss and total loss ablation on 20k pretraining gradient steps. The model trained with total loss converges faster than the one trained with masked loss.

## 5 Conclusion

This paper presents masked decision prediction (MaskDP), a simple and scalable self-supervised method for reinforcement learning (RL) inspired by current large language and vision models. MaskDP is capable of learning scalable and generalizable agents for reinforcement learning that can learn from diverse-quality data sources and infer tasks in goal-reaching and skill-execution settings. Through our empirical study we find MaskDP models outperform past work in zero-shot goal reaching and transfer well to downstream RL tasks, performing competitively with prior pretraining and training from scratch methods.

**Limitations and Future Work**   Computer vision and NLP domains have shown that the true promise of masking architectures lies with their ability to ingest diverse, fully unsupervised data. We study how MaskDP performs when trained without access to expert-level data, and evaluated on unseen proprioceptive states. In the future, we can extend our method to pixel inputs, and pretrain the model to adapt to far different downstream tasks.

The architecture used in MaskDP closely resembles a model-based method, as states are predicted sequentially from actions. In this paper, we use use the predicted next actions directly as this is the simplest and fastest approach. However, it is straightforward to extend MaskDP to plan through our learned model and compare against related baselines.

**Societal Impact**   This is an algorithm for training agents in the style of recent large-scale CV and NLP models. While we do not anticipate particular social risks from our method, as algorithms become capable of ingesting large-scale, in-the-wild data it is important to ensure the dataset does not reinforce undesirable biases or promote harmful behaviors.

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
