# A  Experimental Details

## A.1  Environments and Tasks

We provide the details about the environments and tasks used in our experiments in Table 1.

| Domain | Quadruped | | Walker | | | Cheetah | |
|---|---|---|---|---|---|---|---|
| Task | run | walk | run | walk | stand | run | run_backward |
| State dim | 79 | | 18 | | | 24 | |
| Action dim | 12 | | 6 | | | 6 | |

**Table 1:** Environments and tasks from the DeepMind control suite [29] used in our experiments.

## A.2  Datasets

**Near-Expert Dataset**   We use TD3 agent [11] trained with 1M steps in the above tasks. We then freeze its parameters and rollout its policy with Gaussian noise $N(0, 0.2)$ in every dimension of the action space. For each task, we collect 4M steps of experience (4K episodes in total) using this pretrained TD3 agent.

**Mixed Dataset**   Mixed dataset consists of the following data from various RL agents:

- Near-expert data: Same as the above near-expert dataset, but we only include 2M steps experience (2K episodes in total) for each task.
- Unsupervised data: We use an unsupervised RL algorithm, Proto-RL [36], to collect diverse unsupervised data. We train the agent for 10M steps in each domain and record all the 10M steps (10K episodes).
- Semi-supervised data: We train a TD3 agent to optimize the sum of Proto-RL [36] intrinsic reward and extrinsic reward. The agent is trained for 2M steps in each task and we record all the 2M steps (2K episodes) for each task.

## A.3  Hyperparameters

We provide more details about the hyperparameters and other settings of model training and evaluation in Table 2.

## A.4  Training Details

**Goal-MLP Training**   We adapt the training of Goal-MLP to make it learn to reach goals with varying time budgets. Given a state-action sequence $(s_t, a_t, s_{t+1}, ..., s_{t+m})$, Goal-MLP randomly sample two states $s_i$ and $s_j$ as starting state and goal, (where $t \leq i < j \leq t + m$), and predicts the action $a_i$.

**Goal-GPT Training**   Given a state-action sequence $(s_t, a_t, s_{t+1}, ..., s_{t+m})$, Goal-GPT treats $g = s_{t+m}$ as goal. Every state $s_i$ (where $t \leq i < t + m$) is concatenated with $g$. Then Goal-GPT predicts the action sequence $a_t, ... a_{t+m-1}$ from the state-goal sequence $(s_t, g), ... (s_{t+m-1}, g)$ by passing through causal self-attention layers. In this way, all the goal-reaching baselines are pre-trained to reach goals in various timesteps.

**GPT Training**   Given a state-action sequence $(s_t, a_t, s_{t+1}, ..., s_{t+m})$, GPT predicts the next token (state or action) conditions on previous token sequence, *i.e.*, predicting $s_j$ $(j > t)$ from $s_t, a_t, ... s_{j-1}, a_{j-1}$.

## A.5  Compute Resources

MaskDP is designed to be accessible to the RL research community. The whole pipeline, including data collection, pretraining, and finetuning, only requires a single GPU. All experiments were run on

| MaskDP | Value |
| --- | --- |
| # Context length | 64 |
| # Encoder layer | 3 |
| # Decoder layer | 2 |
| # Attention head | 4 |
| # Hidden dimension | 256 |
| Mask ratio | [0.15, 0.35, 0.55, 0.75, 0.95] |

| GPT/Goal-GPT | Value |
| --- | --- |
| # Context length | 64 |
| # Attention layer | 5 |
| # Attention head | 4 |
| # Hidden dimension | 256 |

| Goal-MLP | Value |
| --- | --- |
| # Context length | 64 |
| # Linear layer | 5 |
| # Hidden dimension | 1024 |

| Training | Value |
| --- | --- |
| Optimizer | Adam |
| $(\beta_1, \beta_2)$ | (.9, .999) |
| Learning rate | $1e^{-4}$ |
| Batch size | 384 |
| # Gradient step | 400000 |

| Evaluation | Value |
| --- | --- |
| # seed | 3 |
| # Goals (single-goal reaching) per seed | 300 |
| # Goals (multi-goal reaching) per seed | $100 \times 5$ |
| Prompt context length | 5 |
| Discount (for RL) | 0.99 |
| Replay buffer size (for RL) | $2M$ |

**Table 2:** Hyperparameters used for model training and evaluation.

GPU clusters with 8 NVIDIA TITAN Xp. The pretraining takes 6-8 hours for 400k gradient steps on the collected datasets using a single GPU.

# B  Additional Experimental Study

## B.1  Dataset Quality

MaskDP has no assumption about the pretraining dataset. To show it doesn't rely on the expert data, we reconstruct another highly diverse dataset called **mixed-v2**, which contains:

- unsupervised data: we train a TD3 [11] agent to maximize Proto-RL [36] intrinsic reward, and store its 10M replay buffer on each domain.

- semi-supervised data: we train a TD3 agent to maximize the sum of extrinsic reward and the Proto-RL intrinsic reward, and store its 2M replay buffer on each task.

- supervised data: we train a TD3 agent to maximize extrinsic reward and store its 2M replay buffer on each task.

The **mixed-v2** dataset is more diverse, as it replaces the near-expert data with TD3 training samples, which are more suboptimal and noisy. After pretraining using **mixed-v2**, we evaluate its performance on unseen state-goal pairs from **near-expert** dataset (dataset in the main paper). So the pretraining and evaluation datasets are in different distributions.

In Figure 12 and Figure 13, we find our model consistently outperforms baselines on all the domains. Compared with Figure 4 and Figure 5, it has more advantages when dataset is noisy, as BC-based methods highly rely on the dataset quality.

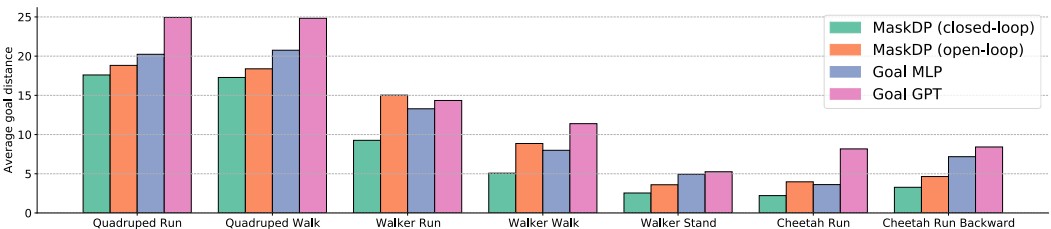

**Figure 12:** The single goal reaching results on **near-expert** goal reaching, after pretraining MaskDP on **mixed-v2** dataset.

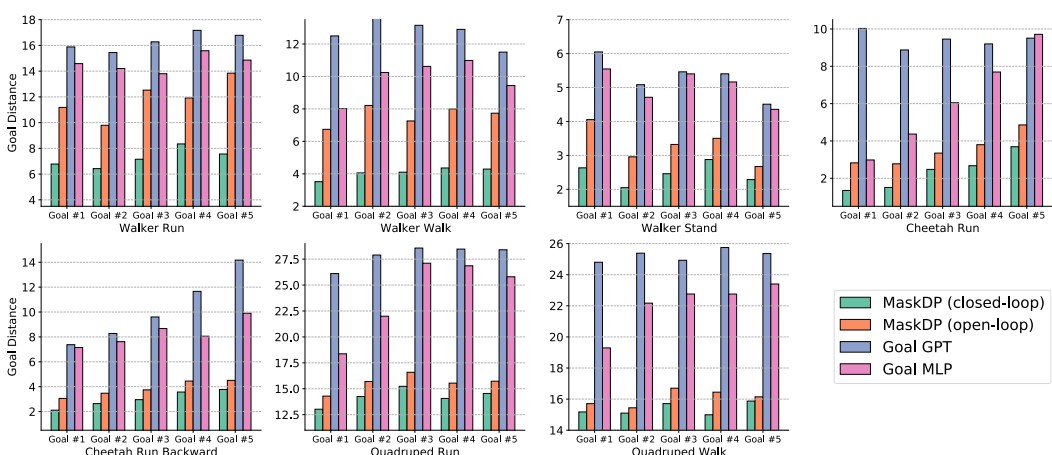

**Figure 13:** The multiple goal reaching results on **near-expert** goal reaching, after pretraining MaskDP on **mixed-v2** dataset.

## B.2 Foresight Helps Multi-goal Reaching

We add ablation about whether to provide multiple future goals to the agent for multi-goal reaching. In contrast, We can also give the agent an individual goal at one time, and switch to a different goal when the budget is exhausted.

## B.3 Trajectory Length Affects Generation Quality

As in [12, 31], we also use sinusoidal positional embedding and perform linear interpolation when the trajectory is longer than the training time. Figure 15 shows the results when we execute the agent for 60, 90, and 120 steps with 5 context tokens, where the training trajectory length is 64. We found on most environments, closed-loop MaskDP can achieve similar performance with GPT and the expert return (the gray bar), except for Cheetah tasks. For longer trajectories, the mask ratio can be extremely low at the beginning, which can cause some bad initial behavior. Meanwhile, GPT can perform stably well as it's not conditioned on masked inputs.

## B.4 Additional Domain: Jaco

We also add a Jaco arm reaching task in the robotics domain. The training and evaluation both follow Section B.1 As shown in Figure 16, MaskDP still outperforms baselines on this task.

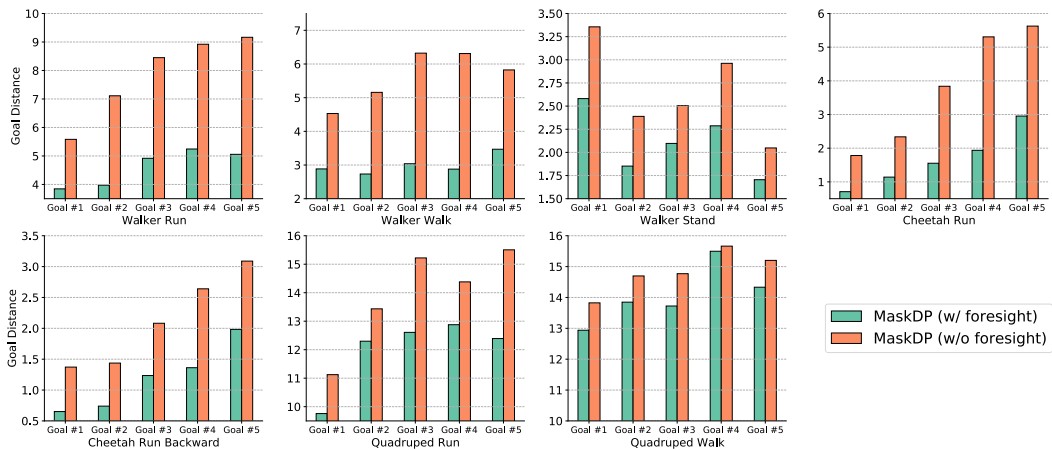

**Figure 14:** We test the closed-loop performance of MaskDP to understand whether the visibility of future goals can improve the performance. We found that on all the domains, MaskDP with foresight performs better.

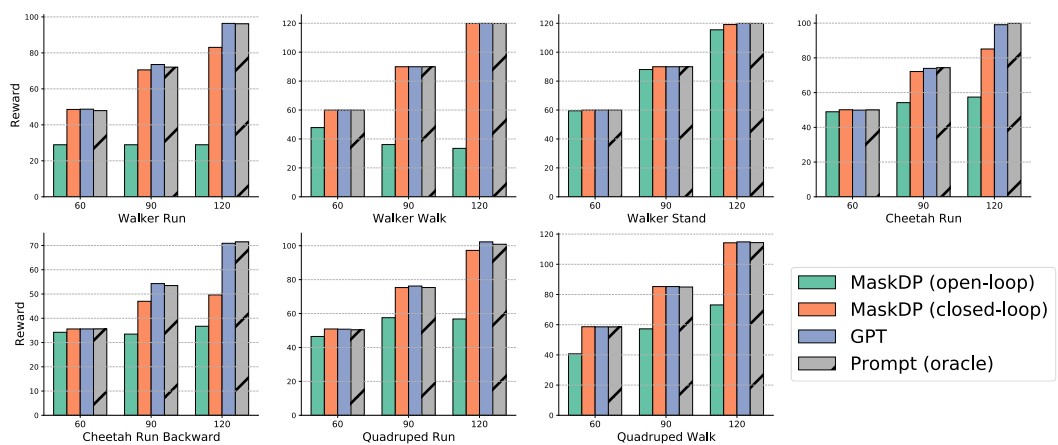

**Figure 15:** Skill prompting performance for longer rollouts.

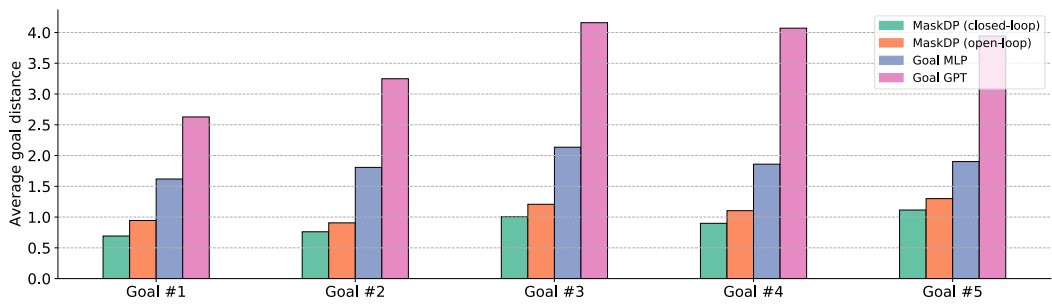

**Figure 16:** Multi-goal reaching results on Jaco reaching task. MaskDP outperforms baselines on this domain.

# C  Full Experimental Results

## C.1  Single-goal Reaching Results

We provide the single-goal reaching results in Table 3 and Table 4 for single-task and multi-task respectively.

| Domain | Task | Goal-MLP | Goal-GPT | Ours (open-loop) | Ours (closed-loop) |
|---|---|---|---|---|---|
| Quadruped | run | 17.832±0.321 | 18.313±0.171 | 13.753±0.255 | 12.912±0.018 |
| | walk | 22.965±0.077 | 23.051±0.055 | 15.456±0.176 | 15.116±0.396 |
| Walker | run | 8.15±0.080 | 9.197±0.012 | 9.565±0.157 | 7.789±0.220 |
| | walk | 5.111±0.118 | 6.029±0.415 | 5.822±0.281 | 2.63±0.158 |
| | stand | 3.979±0.293 | 4.108±0.317 | 3.242±0.1527 | 2.393±0.076 |
| Cheetah | run | 1.377±0.037 | 2.878±0.057 | 2.234±0.145 | 1.385±0.122 |
| | run backward | 1.447±0.136 | 3.011±0.095 | 1.752±0.052 | 0.866±0.065 |

**Table 3:** single-task single-goal reaching results.

| Domain | Task | Goal-MLP | Goal-GPT | Ours (open-loop) | Ours (closed-loop) |
|---|---|---|---|---|---|
| Quadruped | run | 17.825±0.218 | 18.224±0.551 | 13.214±0.230 | 12.926±0.382 |
| | walk | 22.977±0.484 | 23.361±0.201 | 14.892±0.140 | 15.428±0.203 |
| Walker | run | 8.600±0.095 | 8.977±0.181 | 10.242±0.149 | 6.233±0.149 |
| | walk | 5.550±0.170 | 6.083±0.392 | 6.584±0.487 | 4.042±0.148 |
| | stand | 4.096±0.015 | 4.137±0.408 | 3.422±0.161 | 2.248±0.026 |
| Cheetah | run | 1.634±0.042 | 2.953±0.085 | 1.924±0.099 | 1.939±0.125 |
| | run backward | 1.694±0.061 | 2.980±0.076 | 1.378±0.055 | 1.395±0.011 |

**Table 4:** Multi-task single-goal reaching results.

## C.2  Multi-goal Reaching Results

We provide multi-goal reaching results for multi-task pretrained models in Table 5, Table 6, Table 7, Table 8 and Table 9.

| Domain | Task | Goal-MLP | Goal-GPT | Ours (open-loop) | Ours (closed-loop) |
|---|---|---|---|---|---|
| Quadruped | run | 15.644±0.193 | 15.925±0.726 | 10.396±0.152 | 10.213±0.644 |
| | walk | 21.963±0.241 | 22.114±0.445 | 13.413±0.545 | 12.99±0.073 |
| Walker | run | 7.562±0.385 | 7.588±0.243 | 5.981±0.006 | 4.032±0.265 |
| | walk | 5.238±0.212 | 5.481±0.172 | 4.256±0.080 | 2.721±0.213 |
| | stand | 4.295±0.050 | 4.483±0.189 | 2.998±0.271 | 2.293±0.426 |
| Cheetah | run | 1.381±0.151 | 2.342±0.009 | 0.995±0.132 | 0.738±0.041 |
| | run backward | 1.356±0.102 | 2.829±0.058 | 0.811±0.089 | 0.647±0.007 |

**Table 5:** Distance to the first goal in multi-task multi-goal reaching.

| Domain | Task | Goal-MLP | Goal-GPT | Ours (open-loop) | Ours (closed-loop) |
|---|---|---|---|---|---|
| Quadruped | run | 18.183±1.883 | 17.915±0.649 | 11.44±1.065 | 11.736±0.796 |
| | walk | 22.628±1.006 | 23.225±1.325 | 13.986±0.515 | 14.487±0.937 |
| Walker | run | 8.161±0.939 | 8.98±0.775 | 7.165±0.305 | 4.398±0.599 |
| | walk | 5.576±0.778 | 6.458±0.509 | 6.435±0.507 | 2.886±0.217 |
| | stand | 4.389±0.119 | 4.013±0.131 | 2.566±0.508 | 2.045±0.282 |
| Cheetah | run | 1.814±0.126 | 2.75±0.115 | 1.207±0.056 | 1.163±0.032 |
| | run backward | 1.802±0.118 | 3.341±0.180 | 0.917±0.022 | 0.853±0.161 |

**Table 6:** Distance to the second goal in multi-task multi-goal reaching.

| Domain | Task | Goal-MLP | Goal-GPT | Ours (open-loop) | Ours (closed-loop) |
|---|---|---|---|---|---|
| Quadruped | run | 18.377±0.324 | 18.911±1.58 | 12.334±1.26 | 12.085±0.710 |
| | walk | 22.787±0.269 | 23.907±0.946 | 14.7±0.166 | 14.069±0.498 |
| Walker | run | 9.05±0.3809 | 9.178±0.278 | 9.053±0.157 | 5.045±0.179 |
| | walk | 6.127±0.268 | 6.603±0.489 | 5.895±0.097 | 3.113±0.106 |
| | stand | 4.227±0.193 | 4.093±0.225 | 2.878±0.018 | 2.012±0.115 |
| Cheetah | run | 2.329±0.108 | 3.216±0.259 | 1.456±0.085 | 1.519±0.048 |
| | run backward | 2.11±0.114 | 3.807±0.131 | 1.291±0.102 | 1.216±0.026 |

**Table 7:** Distance to the third goal in multi-task multi-goal reaching.

| Domain | Task | Goal-MLP | Goal-GPT | Ours (open-loop) | Ours (closed-loop) |
|---|---|---|---|---|---|
| Quadruped | run | 19.564±0.142 | 19.227±0.156 | 12.92±0.52 | 12.89±0.030 |
| | walk | 23.475±0.342 | 24.119±0.165 | 14.235±0.800 | 14.607±0.698 |
| Walker | run | 8.374±0.536 | 8.926±0.582 | 7.787±0.789 | 4.713±0.751 |
| | walk | 5.548±0.282 | 5.988±0.193 | 6.248±0.402 | 2.884±0.007 |
| | stand | 4.195±0.086 | 4.07±0.103 | 2.704±0.043 | 2.144±0.210 |
| Cheetah | run | 2.501±0.130 | 3.537±0.361 | 1.929±0.034 | 1.971±0.047 |
| | run backward | 2.491±0.047 | 4.145±0.330 | 1.825±0.217 | 1.48±0.166 |

**Table 8:** Distance to the fourth goal in multi-task multi-goal reaching.

| Domain | Task | Goal-MLP | Goal-GPT | Ours (open-loop) | Ours (closed-loop) |
|---|---|---|---|---|---|
| Quadruped | run | 18.749±0.796 | 19.083±1.376 | 13.057±0.202 | 12.162±0.084 |
| | walk | 23.772±0.668 | 24.152±1.002 | 15.275±0.910 | 15.24±1.264 |
| Walker | run | 8.563±0.612 | 8.567±0.196 | 8.955±0.352 | 5.338±0.392 |
| | walk | 6.876±1.103 | 8.334±1.613 | 7.231±1.101 | 3.664±0.276 |
| | stand | 3.93±0.796 | 3.775±0.648 | 2.539±0.394 | 2.009±0.414 |
| Cheetah | run | 3.375±0.456 | 4.623±0.609 | 2.769±0.232 | 2.716±0.335 |
| | run backward | 2.737±0.229 | 4.07±0.164 | 2.244±0.118 | 1.981±0.002 |

**Table 9:** Distance to the fifth goal in multi-task multi-goal reaching.

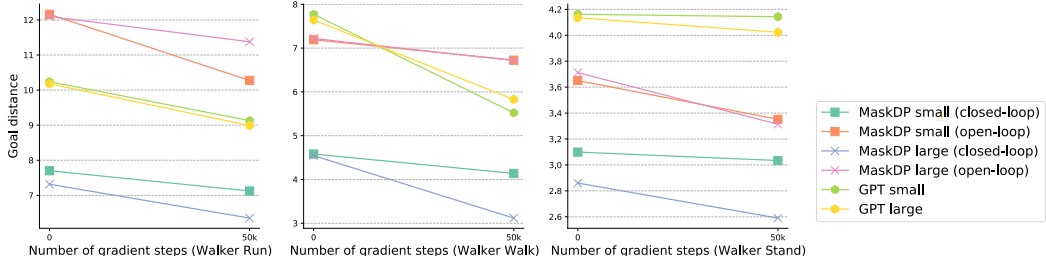

**Figure 17:** Model scalability on walker domain. X-axis represents number of gradient steps. With MaskDP pre-training, larger models outperform smaller models across all Walker tasks. In contrast, Goal-GPT does not have such properties.

## C.3 Finetuning Results on Model Scalability

We pretrain MaskDP using the diverse multi-task mixed dataset, and finetune it using near-expert dataset on each task. In addition to the results in Figure 9 on the Quadruped domain, we also provide results on the other two domains in Figure 17 and Figure 18. Here "small" represents a model with 3 attention layers, while "large" represents 5 attention layers.

We can see the large model with closed-loop evaluation always performs the best, while for Goal-GPT the results are much worse, and the gain from the large model is not significant.

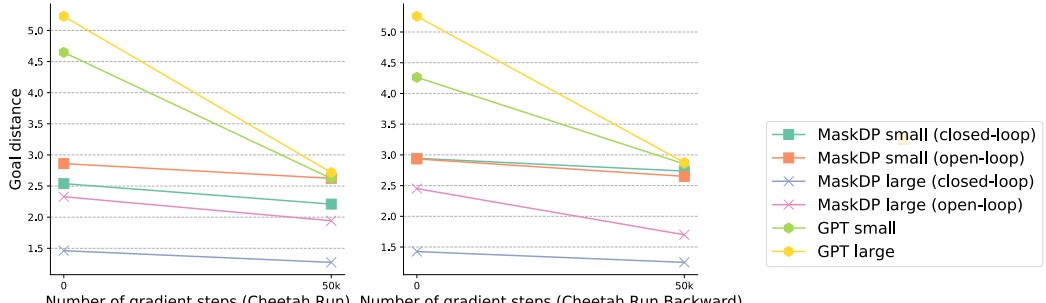

**Figure 18:** Model scalability on cheetah domain. X-axis represents number of gradient steps. With MaskDP pre-training, larger models outperform smaller models across all Walker tasks. In contrast, Goal-GPT does not have such properties.