# OpenReview forum: "Masked Autoencoding for Scalable and Generalizable Decision Making"
_NeurIPS.cc/2022/Conference — NeurIPS 2022 Accept_

### Official Review · Reviewer_cGtJ · 2022-07-11

**Rating:** 8
**Confidence:** 4
**Soundness:** 4 excellent
**Presentation:** 4 excellent
**Contribution:** 4 excellent

**Summary:**

The paper proposes to adapt the Masked Autoencoder (MAE) from computer vision into decision transformer for RL demonstration modeling. The paper propose several downstream application after the unsupervised pre training: Goal reaching, prompting, and fine tuning. The experiments are done in the deep mind suit with Mojuco. The authors also provides analysis on masking ratio and scalability.

**Questions:**

Can you also add experiment on robotics domain like Metaworld?

**Ethics Review Area:**

["I don’t know"]

**Limitations:**

Not that I can see

**Strengths And Weaknesses:**

Strengths:
This paper is well written and easy to follow. While the idea of MAE is not novel, the application of MAE into RL is novel and the proposed downstream prompting and goal reaching are novel, which I think will have large impact. The experiment results are convincing and the choice of baselines are reasonable, showing the effectiveness of mixed masking ratio and versatilty of the large pretrained models.

Weakness:
I couldn’t find any major flaw, but one thing I would love to see is how is this work compared with PromptDT [1]. While I understand these can be concurrent work, but it could be better if there is discussion / comparison between these two very similar papers.

[1] https://arxiv.org/abs/2206.13499

---

> ### Author Response · Authors · 2022-08-02
> **Response to reviewer cGtJ**
>
> We thank the reviewer for the detailed comments and constructive suggestions. We address the questions and concerns below.
>
> > Comparison with PromptDT
>
>
> PromptDT uses trajectory segments as task prompts for multitask learning, which is similar to the GPT baseline in our skill prompting task (Section 4.2.2). We also use a very short segment as task representation on multitask setup, where we showed that a bidirectional transformer like MaskDP can do as well as GPT as a generative model. However, our method doesn’t need specific prompt-based pretraining like PromptDT, but it’s more general and scalable. We just pretrain the model using a simple MAE objective and it can be adapted to goal-reaching, skill/task prompting, RL, etc.
> PromptDT is a newly-released concurrent work (the arXiv link was public after the NeurIPS deadline), and we could not find any official codebase yet. We are glad to discuss it in related work, and compare it in the future when the code is released.
>
> > Robotics Environments
>
>
> Due to the rebuttal time limit, we choose a Jaco arm reaching environment in the DM control suite to reuse our environment wrapper. We include the multi-goal reaching results in Appendix A.5 - Figure 17, where our method outperforms baselines in this domain. We are willing to include more robotics environments in the future.

---

### Official Review · Reviewer_PTCG · 2022-07-11

**Rating:** 7
**Confidence:** 4
**Soundness:** 3 good
**Presentation:** 4 excellent
**Contribution:** 3 good

**Summary:**

This work proposes to use masked autoencoding to learn a self-supervised encoder-decoder transformer model from sampled trajectories. The tokens that are masked and then predicted are a sequence of (interleaved) states and actions. The trained model is then used for goal reaching (both in an open- and closed-loop setting), “skill prompting” and offline RL. Experiments on DM control suite tasks show promising results vs. imitation learning, and an autoregressive baseline.

**Questions:**

- For the goal reaching task, the trajectory length must be fixed in advanced. How is this dealt with in practice? Is there a “skip this step” token?
- How does trajectory length interact with training? Does the model generalize to trajectory lengths it has not seen during training? If not can that be fixed with an appropriate positional encoding, e.g. ALiBi (Press et al., “Train Short, Test Long …”)?
- In the multi-goal setting, how does MaskDP perform if future goals are not available, e.g. if they are unmasked gradually? This baseline should help confirm that foresight is being leveraged to “produce a better plan”.
- For the offline RL task, is it only the encoder that is being finetuned (as typical in vision tasks) or is it both encoder and decoder?

**Limitations:**

I'm concerned that the effect of expert (data-generating) policy on downstream tasks seems a bit too important to be left to future work. If it is left to future work, the authors should include at least some preliminary results in the next revision.

**Strengths And Weaknesses:**

### Positive

- The method is straightforward, and the downstream use of the pre-trained model is crafty. Skill-prompting appears to solve an interesting inference problem. Insights on the appropriate masking strategy in this domain appears useful.
- The experiments are well designed, and the performance is measured online (as opposed to just offline). The baselines are appropriate, and highlight the masking aspect while controlling for confounders.
- The presentation is nice, and the paper is generally well written.

### Not Positive

- The application appears to be somewhat limited in scope (only robotics environments), and it is not clear if and how it might extend to pixels (though a VQGAN like approach can be promising).
- The influence of the policy used to generate pre-training data is not studied but left to future work.

---

> ### Author Response · Authors · 2022-08-02
> **Response to reviewer PTCG**
>
> We thank the reviewer for the detailed comments and constructive suggestions. We address the questions and concerns below.
>
> > Dataset Quality
>
>
> In the main paper, we report results on two datasets: near-expert (section 4.2) and mixed data (Figure 9, Figure 18-19). We want to clarify that our method doesn’t rely on access to expert data. On the mixed dataset (~20% near-expert data with diverse exploratory data), our method can still outperform baselines (see Figure 9, Figure 18-19).
> In order to further address that our method doesn’t make assumptions about dataset quality, we construct a more diverse dataset called mixed-v2, as described in Appendix A.1. This dataset contains few proportions of expert-level experience. We pretrain MaskDP using mixed-v2, and evaluate its performance on the state-goal pairs sampled from near-expert dataset. From Figures 12-13, we can see it outperforms baselines by a large margin. The advantage is more visible than the results in the main paper, as BC-based methods highly rely on the dataset quality.  We also adjust the future work section accordingly, which is marked with blue color.
>
> > Fixed Trajectory length
>
>
> Our method needs to know the time budget for the goal. In this paper, all the methods use the ground-truth time difference plus some additional budgets, and we report the closest distance to the goal. In practice, we can train a time budget estimator and execute it in a close-loop way, where the plan can be more and more confident based on new observations. We leave this extension for future work.
>
> > Trajectory length extrapolation
>
>
> We are using the sinusoid positional embedding, and when the trajectory length is longer than the pretrained length, we use linear interpolation to get longer PE as in MAE [1] and DEiT [2]. the agent is trained with trajectory length 64, and prompted with 5 initial tokens. We report its performance for executing 60, 90, and 120 steps in Appendix A.3 - Figure 15. We found on most environments, close-loop MaskDP can achieve similar performance with GPT and the expert return (the gray bar), except for Cheetah tasks. That’s because the mask ratio can be extremely low at the beginning (5/(120+5)=4%), which can cause some bad initial behavior. It can also address the poor performance of open-loop MaskDP. Meanwhile, GPT can perform stably well with interpolated positional embedding as it’s mask-free.
>
> We believe combining Alibi can further improve length extrapolation and thank the reviewer for the suggestion.
>
> > Foresight vs. Reaching goals one-by-one
>
>
> We provide the results of comparing foresight and one-by-one goal reaching in Appendix A.2 - Figure 14. We can see having foresight actually helps the agent generate a better plan.
>
> > For the offline RL task, is it only the encoder that is being finetuned (as typical in vision tasks), or is it both encoder and decoder?
>
>
> Just like in CV/NLP, we also only finetune the encoder. The trajectory representation will pass through value heads and policy heads to perform RL training. We clarified this point in the updated paper on Page 7.
>
> > Pixel environments
>
>
> We only focus on proprioceptive inputs and leave pixel inputs for future work. Using VQGAN is a promising way of applying our method to pixel input and we appreciate the reviewer’s suggestion.
>
> [1] He et al., Masked autoencoders are scalable vision learners
>
> [2] Touvron et al., Training data-efficient image transformers & distillation through attention

---

> > ### Comment · Reviewer_PTCG · 2022-08-04
> > **Thank you for your response.**
> >
> > I appreciate the clarifications and the additional experiments around foresight.
> >
> > > In order to further address that our method doesn’t make assumptions about dataset quality, we construct a more diverse dataset called mixed-v2, as described in Appendix A.1
> >
> > I appreciate the additional experiments, but I remain half convinced about this. On the one hand, I do expect diverse trajectories generated by an exploration policy trained with e.g. [DIAYN](https://arxiv.org/abs/1802.06070) (or a more recent alternative) to improve the performance of the proposed method. But on the other hand, if the data-generating policy does not adequately expose the environment state space, I can easily imagine that the performance suffers. IMO the cherry on this cake would be if one could (in future work) train an exploration policy in tandem with the MaskDP model. Perhaps this policy could be guided to collect data from parts of the state space that causes the loss (or some other uncertainty metric) of MaskDP to be large.

---

> > > ### Author Response · Authors · 2022-08-07
> > > **Response to reviewer PTCG**
> > >
> > > Thank you for your response. We want to address more about dataset quality.
> > >
> > > > diverse trajectories generated by an exploration policy trained with e.g. [DIAYN](https://arxiv.org/abs/1802.06070) (or a more recent alternative) to improve the performance of the proposed method
> > >
> > > Diverse exploratory data is more challenging than near-expert data. The dataset is generated by a non-stationary stochastic policy maximizing transient intrinsic rewards. Thus, the transitions between states can be very diverse in the dataset, and thus hard to be fitted by the model. BC-based methods suffer from this more than ours (see Figure 9, 12-13, 18-19), as they try to directly memorize the diverse actions. As opposed to these methods, MaskDP implicitly learns forward and inverse dynamics by randomly masking states and actions, which is more robust to the dataset quality.
> > >
> > >
> > >
> > > > the data-generating policy does not adequately expose the environment state space, …the performance suffers
> > >
> > > The near-expert dataset also doesn’t adequately expose the environment state space, but it’s from a high-quality policy. In this case, MaskDP can still outperform baselines (see Figure 2-5).
> > >
> > >
> > >
> > > To summarize, MaskDP can achieve SoTA performance if
> > >
> > > > - the dataset is clean and of relatively high quality (not challenging for BC-based methods)
> > >
> > > > - the dataset is of low quality but can give good coverage of the MDP (more challenging)
> > >
> > >
> > > Here quality means the average return in the dataset. If we couldn’t get datasets in these properties, MaskDP will suffer. Actually, in this case, it’s very challenging for pure offline methods, as the offline data neither tells you what’s a good policy, nor the MDP of the task. As the reviewer suggested, we need to combine offline pretraining with online interactions to actively collect useful data for policy learning. MaskDP-based intrinsic reward is in a spirit similar to ICM [1], which can be interesting for future work.
> > >
> > >
> > > [1] Pathak et al., Curiosity-driven Exploration by Self-supervised Prediction

---

### Official Review · Reviewer_ufWH · 2022-07-12

**Rating:** 7
**Confidence:** 3
**Soundness:** 3 good
**Presentation:** 3 good
**Contribution:** 3 good

**Summary:**

This paper presents a new way of pretraining RL models such that they can be directly applied to Goal Reaching and Skill Prompting tasks, or finetuned for Offline RL. In contrast to previous works that pre-train these models using an autoregressive objective, the authors propose using the masked autoencoder (MAE) objective.  The bidirectional transformer model is pre-trained through reconstructing masked state and action tokens.  The authors demonstrate how such a model can be deployed to the three types of downstream tasks using distinct masking strategies.  Although the method is a simple extension of MAE pre-training, which has recently shown state-of-the-art performance in certain CV and NLP tasks, this paper shows extensive empirical comparisons against baselines.  In particular, offline RL results on Walker are comparable to SoTA in ExoRL [31].  Ablation study is also provided to justify some of their design choices such as the mixed mask ratio strategy.

[31] D. Yarats, D. Brandfonbrener, H. Liu, M. Laskin, P. Abbeel, A. Lazaric, and L. Pinto. Don’t
380 change the algorithm, change the data: Exploratory data for offline reinforcement learning.

**Questions:**

I wonder if you could replace the word “zero-shot” performance with another word since goal reaching tasks are instances of the pre-training masked autoencoding task.


**Limitations:**

Yes, the authors have addressed the potential broader impact.


**Strengths And Weaknesses:**

Strengths:  The biggest strength of this paper is to investigate the use of MAE for unsupervised pretraining in RL.  Although I might have missed related works, this seems to be the first work that provides practical implementations of the method and comparisons against relevant GPT baselines that use the next state/action prediction loss.  Extensive experiments do show MaskDP outperform autoregressive baselines.
Overall, the paper is clearly written.  It is interesting to see how MAE performs in unsupervised RL given its success in CV and NLP domains.

Weaknesses: As acknowledged by the authors themselves, MaskDP still uses some expert data, which are not “unlabeled” as in CV and NLP settings.   This is okay as the method can be compared against other methods using demonstrations.  Unfortunately, I’m not very aware of the state-of-the-art results on these tasks using demonstrations.  Although the walker result is compared against SoTA, I cannot comment on the significance of this offline RL result section.  Is there a reason why only the Walker domain is compared against SoTA results?

-----
As the authors addressed my concerns, I've updated my score.

---

> ### Author Response · Authors · 2022-08-02
> **Response to reviewer ufWH**
>
> We thank the reviewer for the detailed comments and constructive suggestions. We address the questions and concerns below.
>
> > Dataset Quality
>
>
> In the main paper, we report results on two datasets: near-expert (section 4.2) and mixed data (Figure 9, Figure 18-19). We want to clarify that our method doesn’t rely on access to expert data. On the mixed dataset (~20% near-expert data with diverse exploratory data), our method can still outperform baselines (see Figure 9, Figure 18-19).
> In order to further address that our method doesn’t make assumptions about dataset quality, we construct a more diverse dataset called mixed-v2, as described in Appendix A.1. This dataset contains few proportions of expert-level experience. We pretrain MaskDP using mixed-v2, and evaluate its performance on the state-goal pairs sampled from near-expert dataset. From Figure 12-13, we can see it outperforms baselines by a large margin. The advantage is more visible than the results in the main paper, as BC-based methods highly rely on the dataset quality. We also adjust the future work section accordingly, which is marked with blue color.
>
> > Whether baselines are SoTA on goal-reaching tasks with demonstrations
>
>
> BC generally performs the best on diverse and high-quality datasets, so we study commonly-used architectures for BC (MLP or GPT) and use them for comparison. The Goal-GPT is also a special case of decision transformer[1] by replacing the reward-to-go token with a goal token, so the baselines we compared are at the SoTA level.
>
> > Is there a reason why only the Walker domain is compared against SoTA results?
>
>
> In the ExORL[2] paper, they didn't provide Quadruped/Cheetah results, so we only reported results for Walker and found their performances matched. Due to the dataset quality, we found finetuning using ProtoRL data on Quadruped/Cheetah is not successful for both ExoRL and MaskDP, so we report the RL results finetuned on the TD3 replay (namely “supervised data”) in Appendix A.1. From Figure 16, we can see MaskDP’s performance can match GPT-style pretraining. On proprioceptive inputs, both GPT and MaskDP can learn useful trajectory representation for offline RL, which may not hold on pixel inputs. We leave this part as future work to explore.
>
> > Clarifying zero-shot evaluation
>
>
> Here “zero-shot” means we directly evaluate the model on some unseen state-goal pairs in the validation set (in contrast to “finetuning”). We have clarified this point in the updated paper at Page 5.
>
> [1] Chen et al., Decision Transformer: Reinforcement Learning via Sequence Modeling
>
> [2] Yarats et al., Don’t change the algorithm, change the data: Exploratory data for offline reinforcement learning.

---

### Author Response · Authors · 2022-08-02
**Author response**

We thank all the reviewers for their encouraging comments. We are glad to see that the reviewers generally appreciate our results, novelty and contribution, and think the paper “will have large impact” (cGtJ), the presentation is excellent (ufWH, PTCG, cGtJ), and the solution is novel: “a new way of pretraining RL models such that they can be directly applied to goal reaching, skill prompting and finetuned to offline RL” (ufWH), “the downstream use of the pre-trained model is crafty” (PTCG), “application of MAE into RL is novel and the proposed downstream prompting and goal reaching are novel” (cGtJ).

We address the concerns each reviewer has in the individual responses below.

---

### Meta-Review · Area_Chair_ZVSb · 2022-08-27

**Recommendation:** Accept
**Confidence:** Certain

**Metareview:**

The paper investigates the use of Masked Auto Encoders (MAE) for unsupervised pretraining in RL. Reviewer ufWH summarizes well, "this paper shows extensive empirical comparisons against baselines. In particular, offline RL results on Walker are comparable to SoTA in ExoRL [31]". cGtJ says, "The paper propose several downstream application after the unsupervised pre training: Goal reaching, prompting, and fine tuning. The experiments are done in the deep mind suit with Mojuco. The authors also provides analysis on masking ratio and scalability."

Overall, reviewers find this to be a simple but strong method for unsupervised training for RL. I agree.

**Award:**

No

---

### Decision · Program_Chairs · 2022-09-14

Accept